# Acid sphingomyelinase is a gatekeeper of placental labyrinthine architecture and function

Isidora Rovic[1,2], Katherine Szelag[1,2], Han Li[2], Rosanne McQuaid[2], Sara Sugin[1,2], Que Wu[1], Natalia Theodora[3], John Sled[3,4,5,6] and Andrea Jurisicova[1,2,6,*]

## ABSTRACT

Sphingolipids are a class of bioactive signaling lipids that regulate an array of fundamental cellular processes, including cell survival, proliferation and differentiation. Deficiency of acid sphingomyelinase – an enzyme of the sphingolipid metabolic pathway – has been previously implicated in human placental pathologies. We demonstrate that acid sphingomyelinase (*Smpd1*) is required for normal placental development in the mouse, and its deficiency results in an intrauterine growth restriction phenotype. Smpd1-deficient placentas display several anatomical abnormalities, including a reduced labyrinth compartment and increased fetal-maternal interhaemal distance. Finally, we observed several hallmarks of defective autophagy and lysosomal impairment in *Smpd1*$^{-/-}$ placentas, which could explain the inability of *Smpd1*$^{-/-}$ trophoblast to respond to nutrient starvation. Fetal growth restriction could not be rescued by transfer of *Smpd1*-deficient embryos into a wild-type uterine environment; however, restoration of transcription factor EB phosphorylation was detected. Thus, we conclude that, due to a smaller labyrinthine area, *Smpd1* deficiency leads to a decrease in exchange between maternal and fetal blood space, limiting the supply of nutrients to the fetus and resulting in growth restriction.

KEY WORDS: Placentation, Phospholipids, Acid sphingomyelinase, Lysosome, Trophoblast differentiation

## INTRODUCTION

The placenta is an essential organ required for the proper growth and survival of the fetus. Its development involves complex interactions of numerous cell types, on both maternal and fetal sides. Given its transient nature, the placenta undergoes extensive remodeling during its lifespan, which is regulated by a delicate balance of trophoblast cell proliferation, differentiation and death (Jurisicova et al., 2005). Defects in these processes can lead to placental abnormalities, and, as a result, compromised growth and survival of the fetus. Indeed, placental defects are associated with two of the most common and

serious disorders of human pregnancy: maternal pre-eclampsia and fetal/intrauterine growth restriction. Importantly, heightened trophoblast cell death and improper syncytialization have been observed in placentas from pregnancies complicated either by pre-eclampsia (DiFederico et al., 1999; Roland et al., 2016) or intra-uterine growth restriction (Shao et al., 2021). Thus, dissecting molecular disturbances that underpin altered placental cellular phenotypes observed in pregnancy complications is key to understanding placental pathologies. Although these disorders have been studied for decades, their etiology is still poorly understood.

Studies of mouse placentation have been used as a model to explore defects observed in human pregnancies. The murine placenta becomes fully functional at ∼9.5 days post-coitum (dpc), when it forms three distinct cellular regions: the labyrinth, the junctional zone and the decidua (Rossant and Cross, 2001; Cross, 2005). Chorioallantoic attachment facilitates labyrinth formation, which is responsible for the maternal/fetal exchange of nutrients and waste. The junctional zone provides structural and hormonal support to the labyrinth and to maternal blood vessels. It is demarcated from the mostly maternal decidua by a fetal-derived population of trophoblast giant cells (TGCs). Based on their location and function within the placenta, there are four different TGC subtypes: sinusoidal, parietal, endovascular and mural (Simmons and Cross, 2005). These cells play a role in embryo implantation, trophoblast invasion and pregnancy maintenance (Cross et al., 2002). While the placenta develops intimately with the maternal uterine wall, it is an embryonic structure.

Recent work on pregnancy-associated disorders focused on dysregulated sphingolipid metabolism in both maternal and fetal compartments, and implicated an altered ceramide/S1P rheostat in numerous physiological pregnancy-associated complications (reviewed by Fakhr et al., 2021). Sphingolipids are a distinct class of lipids that are major structural components of all eukaryotic cell membranes. In addition to their structural roles within membranes, several sphingolipid metabolites act as bioactive signaling molecules that regulate a variety of important cellular processes. Ceramide has long been established as a pro-apoptotic signaling molecule and second messenger, while sphingosine-1-phosphate (S1P) promotes cell survival and proliferation (Bartke and Hannun, 2009). Sphingomyelin-phosphodiesterase 1 (SMPD1), also known as acid sphingomyelinase (ASM), is a lysosomal sphingomyelinase responsible for the hydrolysis of sphingomyelin (SM) – a major component of all cellular membranes – into ceramide. When its function is impaired, SM accumulates in the cellular membranes, altering its fluidity. *Smpd1* is well known for its role in the cellular stress response, but it also plays an important role in regulating cellular endolysosomal pathways in several human metabolic disorders (Breiden and Sandhoff, 2021).

An analysis of sphingolipid regulation in placentas from human pregnancies complicated by pre-eclampsia revealed significantly decreased levels of SMPD1 enzymatic activity (Melland-Smith

[1]Department of Physiology, Temerty Faculty of Medicine, University of Toronto, Toronto, Ontario M5S 1A8, Canada. [2]Lunenfeld-Tanenbaum Research Institute, Sinai Health System, Toronto, Ontario M5G 1X5, Canada. [3]Mouse Imaging Centre, 25 Orde Street, Toronto, Ontario M5T 3H7, Canada. [4]Translational Medicine, Hospital for Sick Children, Toronto, Ontario M5G 0A4, Canada. [5]Department of Medical Biophysics, University Health Network-Toronto General Hospital, Toronto, Ontario M5G 1L7, Canada. [6]Department of Obstetrics and Gynecology, Temerty Faculty of Medicine, University of Toronto, Toronto, Ontario M5G 1E2, Canada.

*Author for correspondence ( jurisicova@lunenfeld.ca)

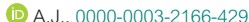 A.J., 0000-0003-2166-4298

et al., 2015). In addition, altered sphingolipid metabolites in placental disorders were also reported (Baig et al., 2013; Melland-Smith et al., 2015). Yet the precise causative mechanisms remain elusive. In this study, we demonstrate that *Smdp1* is required for normal placentation in mice and that its deficiency results in impaired fetal growth. Phospholipidomics revealed a marked reduction in S1P-associated with diminished sphingosine kinase activity. Moreover, we demonstrate that a deficiency of *Smpd1* results in abnormal differentiation of labyrinthine placental compartments with the accumulation of dysfunctional lysosomes and inefficient autophagy. These pathologies cannot be reversed by the transfer of mutant embryos into a normal uterine environment. We therefore conclude that altered placental embryonic sphingolipid metabolism triggered by *Smpd1* deficiency compromises embryonic growth.

## RESULTS
### Smpd1 is expressed throughout placental development and in all trophoblast cell types

Despite the evidence that sphingolipid metabolizing enzymes play a role in embryo development and pregnancy, the expression and activity of murine Smpd1 throughout placental development remains

unknown. We therefore determined the expression, protein localization and enzymatic activity of SMPD1 throughout normal placental development.

There was no difference in *Smpd1* transcript accumulation in placenta during gestation (Fig. 1A). Lack of specific SMPD1 antibodies prompted us to develop a Flag-tagged knock-in version of *Smpd1*. Homozygous *Smpd1[flg/flg]* mice are viable and fertile, and do not develop any obvious phenotypes associated with *Smpd1* deficiency, indicating that the N-terminal tagged version of *Smpd1* retains its function. Tagged SMPD1 migrates at ~75 kDa, corresponding to the size of the precursor SMPD1 protein (Fig. 1B). In the mature lysosomal form (~70 kDa), the N-terminal region becomes cleaved and would thus not be detectable with a Flag antibody. We have observed an increase in SMPD1 protein levels at gestational day (d) 13.5, with a gradual decline towards the end of gestation. However, no change in SMPD1 enzyme activity was observed between d13.5-d17.5 placental lysates (Fig. 1C). We thus conclude that Smpd1 activity in murine placenta is relatively stable throughout gestation.

Finally, we analyzed the spatial distribution of SMPD1 protein via immunohistochemistry (Fig. 1D). Epitope-specific staining was

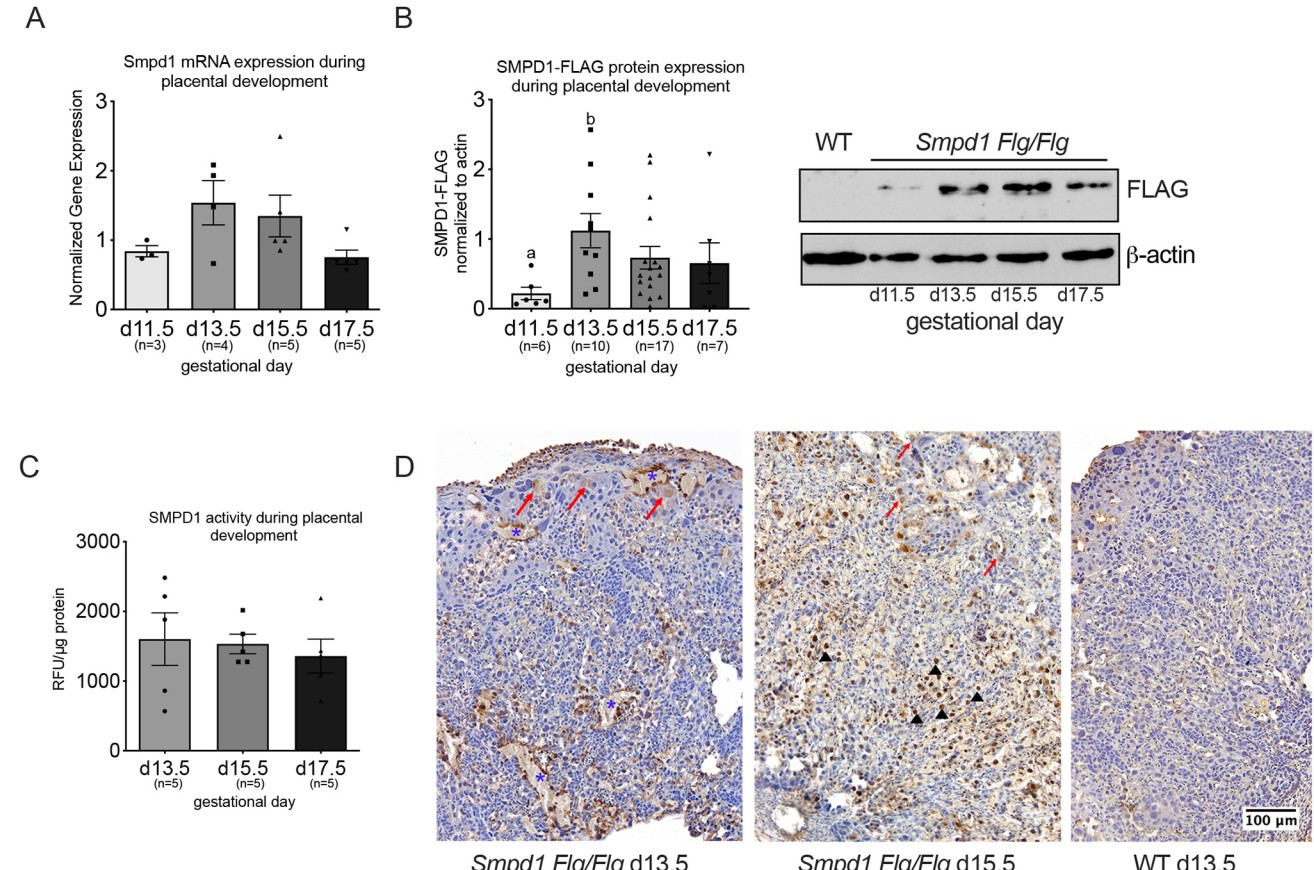

**Fig. 1. Smpd1 expression during normal placental development.** (A) *Smpd1* mRNA expression did not change significantly between gestational day (d) 11.5, d13.5, d15.5 and d17.5 of placental development (Kruskal-Wallis one-way ANOVA and Dunn's test). (B) (Left) Densitometric analysis of Smpd1-FLAG protein expression normalized to β-actin. (Right) Representative immunoblot for SMPD1 (FLAG) expression of FLAG-positive (d11.5-d17.5 aged placenta) and wild-type (d17.5) placenta, where a FLAG-specific band was detected at 75 kDa. *n* indicates the number of placentas that were obtained from three to five different dams (Brown-Forsythe and Welch ANOVA tests and Dunnetts multiple comparisons). (C) SMPD1 enzyme activity did not change between d13.5, d15.5 and d17.5 placentas. *n* indicates the number of placentas obtained from at least two dams (ordinary one-way ANOVA and Tukey's multiple comparisons). (D) Representative immunohistochemical images of *Smpd1 flg/flg* at wild-type placental mid-line sections at d13.5 and d15.5 stained using anti-DYKDDDDK antibody. Cell types showing immunoreactivity are indicated as follows: asterisks indicate canal/spiral arteries lined by fetal giant cells; arrows indicate a patchy signal in spongiotrophoblast; arrowheads indicate some sinusoidal giant cells. All data are mean±s.e.m., the different letters represent statistical significance (*P*<0.05). For details of the statistical test(s) used, see the Materials and Methods.

localized to the junctional zone and labyrinth layer. Specifically in fetal tissues, strong immunoreactivity was present in canal and/or spiral arteries, and subset of sinusoidal trophoblast giant cells within the labyrinth, with occasional patchy signal in spongiotrophopblast cells. Low to no reactivity was observed in wild-type placentas.

### *Smpd1*-deficient fetuses exhibit intrauterine growth restriction and abnormal placental phenotype

*Smpd1*$^{-/-}$ fetuses exhibited a progressive decline in weight, starting at d17.5 ($P<0.05$), that became even more obvious at birth (Fig. 2A). Although fetal weights were reduced, there was no significant difference in placental weights (Fig. S1B). In addition, while there was no difference between wild-type and *Smpd1*$^{-/-}$ litter size at birth (Fig. 2B), there was a significantly increased number of resorptions in *Smpd1*$^{-/-}$ dams (Fig. 2C). This discrepancy could be explained by an increased number of pre-implantation embryos recovered at d3.5 from *Smpd1*-deficient dams (Fig. 2D). Intriguingly, at postnatal day 1 (P1), most of the differences in the surviving pup weights could be attributed to females (Fig. S1B).

The presence of an intrauterine growth restriction phenotype in *Smpd1*$^{-/-}$ fetuses indicated that there may be developmental defects in the placenta. Midline sections of day 17.5 wild-type and *Smpd1*$^{-/-}$ placentas (Fig. 3A) were analyzed to determine the distribution and relative proportions of the decidua, spongiotrophoblast and labyrinth regions. While the total area of wild-type and *Smpd1*$^{-/-}$ placentas was unchanged, *Smpd1*$^{-/-}$ placentas exhibited a greater proportion of junctional zone and a decreased labyrinth (Fig. 3B). However, within the junctional zone,

morphometric analysis did not reveal any differences in the proportion of spongiotrophoblast/glycogen cell compartments. Upon further inspection, the labyrinth region of *Smpd1*$^{-/-}$ placentas showed striking morphological abnormalities. Notably, the interhaemal membrane, which separates fetal capillaries and maternal blood spaces, was thicker in the *Smpd1*$^{-/-}$ labyrinth (Fig. 3C). The maternal spaces that were measured were significantly reduced in both d15.5 and d17.5 *Smpd1*$^{-/-}$ placentas (Fig. 3D). The interhaemal region contains two layers of syncytiotrophoblast cells, ST-I and ST-II. ST-I resides adjacent to the maternal blood spaces (MBSs), and ST-II faces the fetal capillaries (FCs). This barrier is a key place where maternal/fetal exchange occurs and its architecture has impact on the rate of nutrient transfer. Transmission electron micrographs of labyrinth sections revealed that both ST-I and ST-II appeared to be thicker in mutant placentas, and electron-dense inclusions accumulated within the fetal-facing ST-II (Fig. 3E). Interhaemal thickness was significantly increased in the KO labyrinth (Fig. 3F). To further elucidate which ST layer might be most affected by *Smpd1* deficiency, placenta sections were stained for the syncytiotrophoblast (ST) markers MCT1 and MCT4 (Fig. 3G). MCT1 is localized to the apical membrane of ST-I cells facing maternal blood spaces, and MCT4 is localized to the basal membrane of ST-II cells facing fetal capillaries. In comparison to the thin and tightly juxtaposed pattern of MCT1/MCT4 staining observed in the wild-type labyrinth, the labyrinth of *Smpd1*$^{-/-}$ placentas was noticeably diffuse, with a disorganized MCT1 and MCT4 staining pattern. Moreover, the *Smpd1*$^{-/-}$ labyrinth

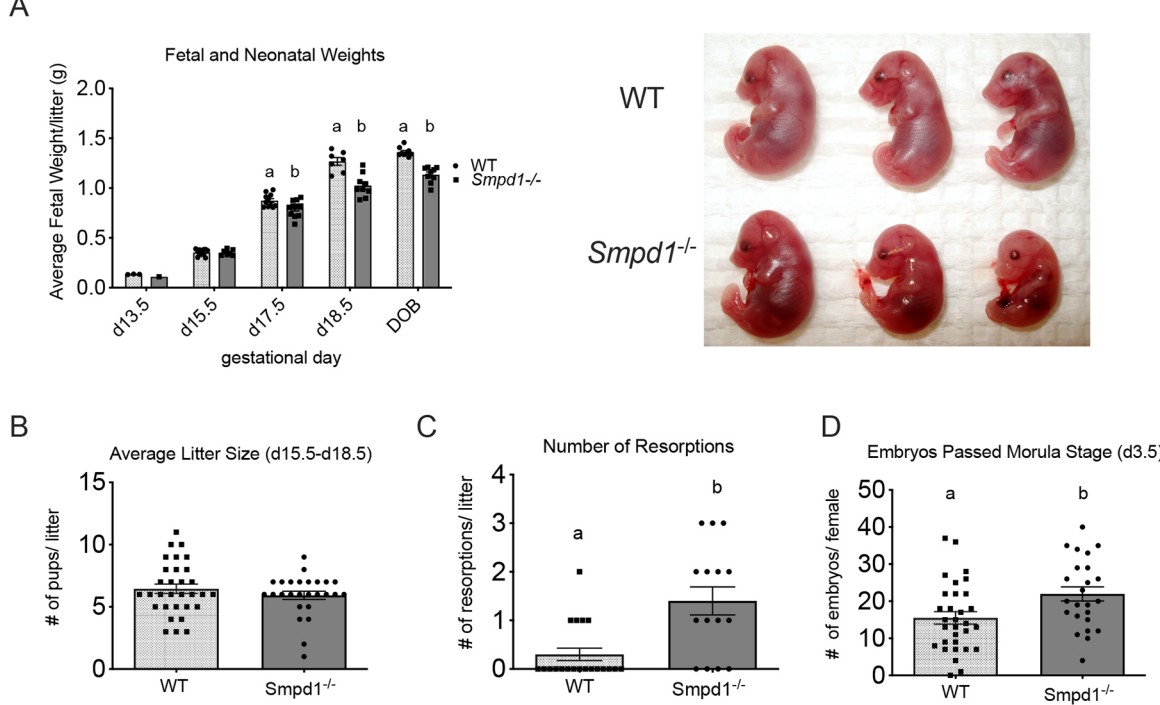

**Fig. 2. Fetal growth restriction and resorbing fetuses in *Smpd1*$^{-/-}$ mice.** (A) (Left) *Smpd1*$^{-/-}$ fetuses exhibit reduced birth weights, starting at day 17.5 of gestation (d17.5). This trend persists until d18.5 and in neonates measured on their day of birth (DOB) (two-way ANOVA and Bonferroni post-hoc tests). (Right) Representative images of the wild-type and *Smpd1*$^{-/-}$ fetal size at d18.5 of gestation. (B) There is no difference in the average litter size (number of pups per litter) between wild-type and *Smpd1*$^{-/-}$ litters (unpaired *t*-test). (C) *Smpd1*$^{-/-}$ litters have a significantly higher number of resorbing fetuses compared with wild type. Data were collected for all litters from age-matched wild-type and *Smpd1*$^{-/-}$ dams (8-10 weeks of age) (Mann-Whitney test). (D) Upon hormonal priming, there is a greater number of d3.5 *Smpd1*$^{-/-}$ embryos produced per female compared with wild type. Statistical significance was assessed using an unpaired *t*-test. The individual datapoints represent the number of litters/dams. All data are mean±s.e.m.; the different letters represent statistical significance ($P<0.05$). For details of the statistical test(s) used, see the Materials and Methods.

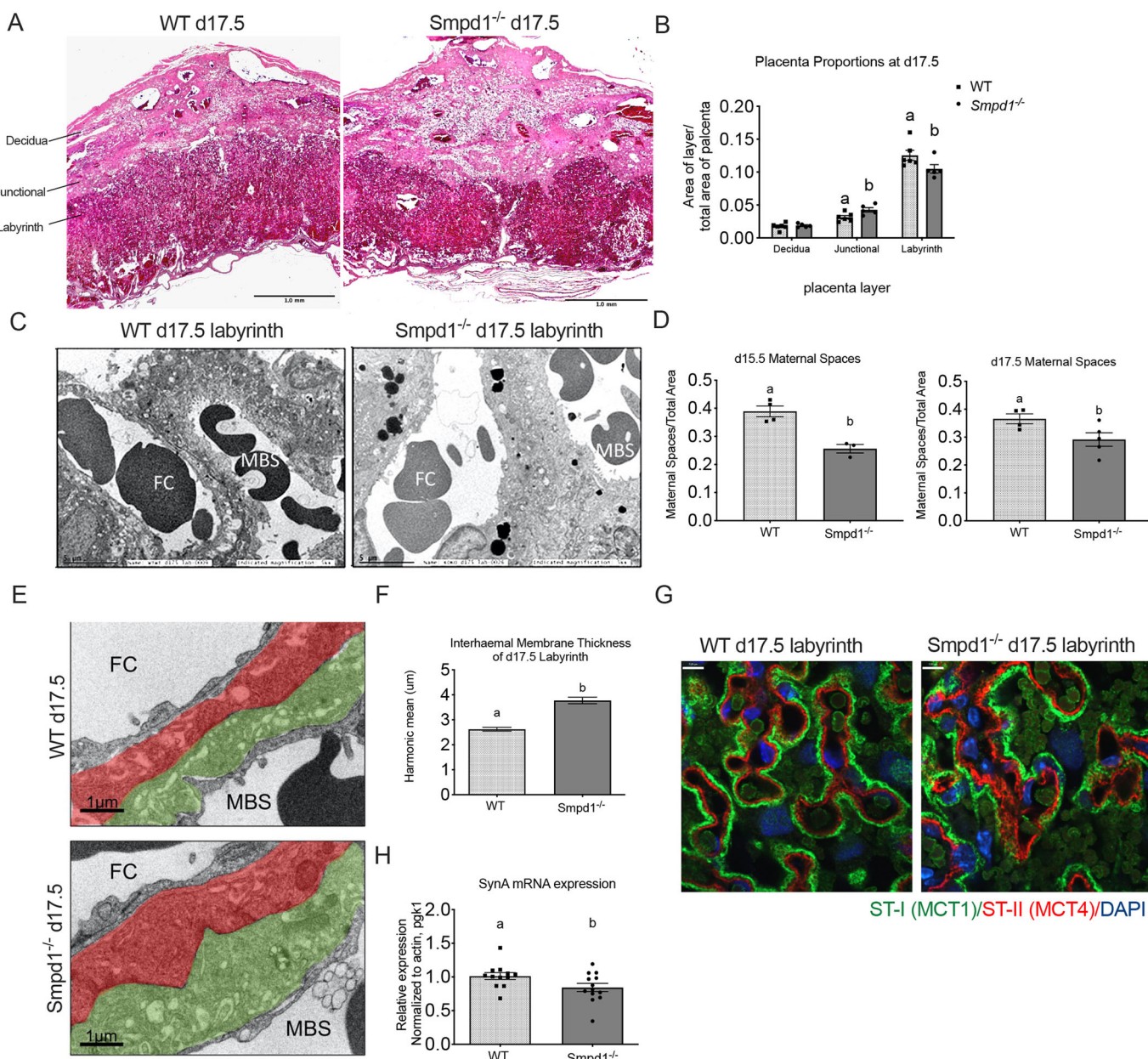

**Fig. 3. Placental defects in *Smpd1*⁻/⁻ mice.** (A) Morphology of gestational day (d) 17.5 wild-type (left) and *Smpd1*⁻/⁻ placenta (right) at 2× magnification. (B) Total placental surface area was not significantly different between wild type and *Smpd1*⁻/⁻; however, the proportion of the junctional layer was increased and the proportion of labyrinth (Lab) was decreased in *Smpd1*⁻/⁻ placentas (*P*<0.05; two-way ANOVA and Bonferroni post-hoc test). (C) Transmission electron micrographs of d17.5 wild-type and *Smpd1*⁻/⁻ placenta interhemal membrane. (D) Maternal membrane space measurements are decreased in *Smpd1*⁻/⁻ d15.5 and d17.5 placentas from six randomized rectangular sections (unpaired *t*-test). (E) Pseudo-colored transmission electron micrographs of d17.5 wild-type (top) and KO (bottom) interhemal membranes. MBS, maternal blood space; FC, fetal capillary; green pseudo-color indicates syncytiotrophoblast layer I (MBS facing); red pseudo-color indicates syncytiotrophoblast layer II (FC facing). (F) Interhemal thickness was significantly increased in KO labyrinth (unpaired *t*-test). (G) Expression of Mct1 (ST-I, green) and Mct4 (ST-II, red) immunofluorescence confocal microscopy in d17.5 wild-type and *Smpd1*⁻/⁻ placentas. Mutant placentas exhibit focally elevated MCT1, as well as less defined and disorganized MCT4 staining. Nuclei are counterstained with DAPI (blue). (H) mRNA levels of *Syna* were significantly decreased in *Smpd1*⁻/⁻ placentas (unpaired *t*-test). Individual data points represent individual placentas, obtained from at least two different dams. Letters indicate statistical significance (*P*<0.05). All data are mean±s.e.m. For details of the statistical test(s) used, see the Materials and Methods. Scale bars: 5 mm in A; 5 µm in C; 1 µm in E; 7 µm in G.

contained regions of increased or thickened MCT1/MCT4 immunoreactivity, indicating either undifferentiated or improperly fused syncytial cells. This was also consistent with decreased expression of fusogenic driver *SynA* encoded by endogenous retrovirus envelope member 1 (Fig. 3H).

Analysis of placental vasculature did not reveal differences in the surface area, volume, diameter, distribution and span of fetal vascular tree (Fig. S2A). However, there was a significant reduction in the average depth to which fetal vessels formed within the mutants (Fig. S2B), a finding that is consistent with the

reduction of the labyrinthine compartment we observed during histomorphometry analysis.

## Elevated sphingomyelin levels and decreased S1P are the most obvious sphingolipids altered in *Smpd1*⁻/⁻ placentas

To determine to what extent sphingolipid metabolism is affected by *Smpd1* placental deficiency, we investigated their phospholipid profile. As expected, tandem mass spectrometry (MS/MS) analysis revealed that sphingomyelin (SM) levels were increased in *Smpd1*⁻/⁻ placentas. Specifically, SM species with a 12-, 16- and 18-carbon fatty acyl-CoA chain were significantly elevated, while the remaining SM species were unaltered (Fig. S2B). Previous studies proposed that SM accumulation within *Smpd1*⁻/⁻ tissues is cell-type specific due to increased activity of compensatory pathways (Lozano et al., 2001). Other than exhibiting different kinetic properties within cellular membranes, the roles of individual SM species in cellular physiology are still largely unknown.

Consistent with SM accumulation, levels of ceramide C16, C24, C24:1 in *Smpd1*⁻/⁻ placentas significantly decreased. Interestingly, levels of sphinganine (Spa) and sphingosine (Sph), two structurally similar phospholipids of distinct metabolic origin (Gault et al., 2010), were elevated, while sphingosine-1-phosphate (S1P) levels decreased ∼3-fold in *Smpd1*⁻/⁻ placentas. This suggested inefficient

conversion of Sph into S1P by sphingosine kinase (SPHK). Indeed, decreased sphingosine kinase activity was detected in *Smpd1*-deficient placental lysates (Fig. S2C). Thus, as expected, *Smpd1* deficiency leads to accumulation of sphingomyelin but unexpectedly revealed decreased levels of S1P in the placenta.

## Apoptotic cell death is not affected in *Smpd1*-deficient placentas

Given the altered levels of sphingolipid metabolites in *Smpd1* KO placentas, many of which have been shown to regulate cell death (Iessi et al., 2020), we wanted to investigate whether cell death patterns (Detmar et al., 2019) were affected by Smpd1 deficiency. While TUNEL-positive nuclei were observed in all regions at both d15.5 and d17.5, no significant difference in the number of dead cells was observed in any region between wild-type and *Smpd1*⁻/⁻ placentas (Fig. S3).

## *Smpd1*⁻/⁻ placentas exhibit impaired autophagy and abnormal lysosomal morphology

Acid sphingomyelinase deficiency is a causative enzymatic defect driving lysosomal dysfunction in the recessively inherited form of Niemann-Pick disease. Given that acid sphingomyelinase is a

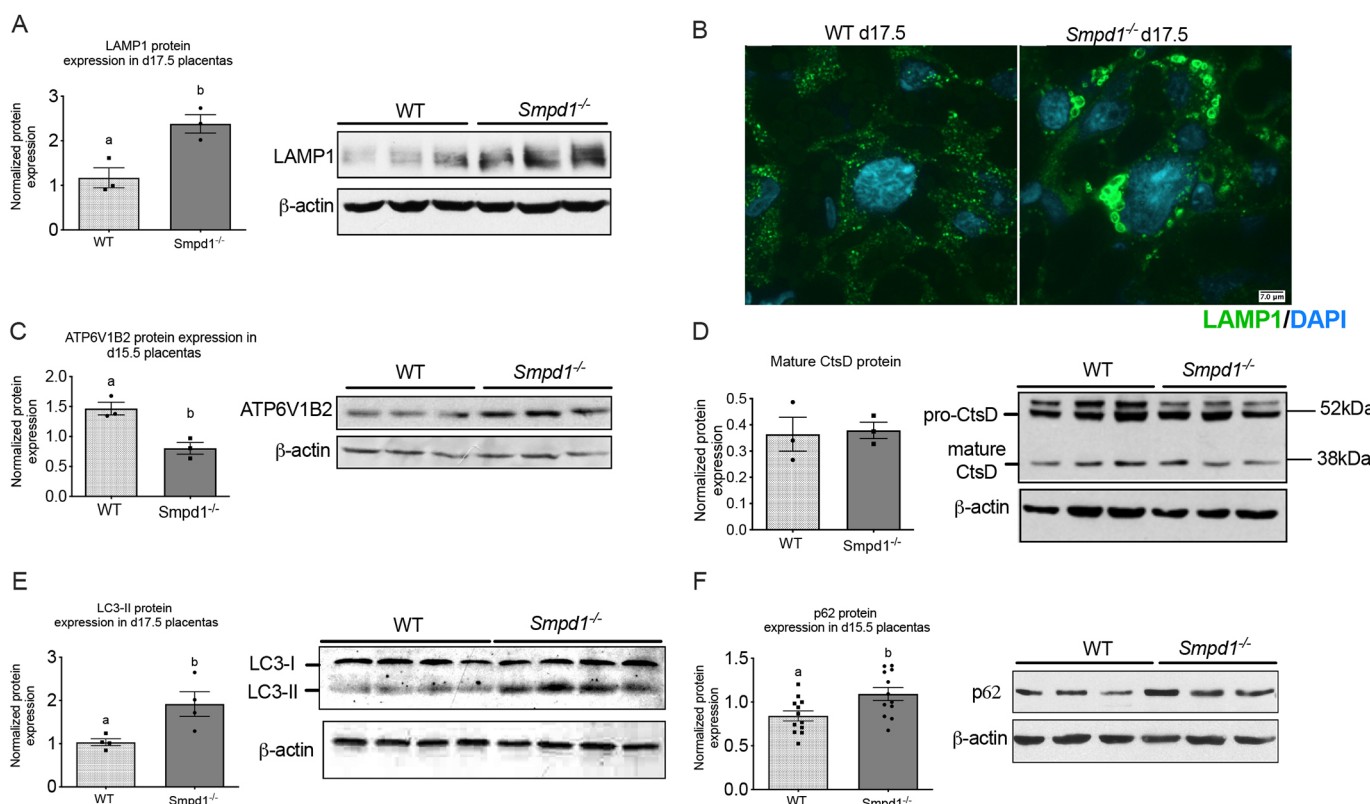

**Fig. 4. Accumulation of lysosomes and autophagosomes in *Smpd1*⁻/⁻ placentas.** (A) Densiometric analysis of LAMP1 protein normalized to β-actin (left) and a LAMP1 immunoblot of wild-type and *Smpd1*⁻/⁻ gestational day (d) 17.5 placentas. LAMP1 protein levels are significantly increased in *Smpd1*⁻/⁻ placentas and, due to its variable glycosylation pattern, appear as a wide band. (B) LAMP1 immunofluorescence staining on d17.5 wild-type and *Smpd1*⁻/⁻ placentas. *Smpd1*⁻/⁻ labyrinthine trophoblast has a greater number and increased size of lysosomes compared to wild type. Scale bar: 5 μm. (C) Densiometric analysis of ATPV1B2 protein expression normalized to β-actin (left) and an ATP6V1B2 immunoblot of wild-type and *Smpd1*⁻/⁻ d15.5 placentas (right). ATPV1B2 protein levels are significantly decreased in *Smpd1*⁻/⁻ placentas ($P<0.01$). (D) Densiometric analysis of CtsD protein expression normalized to β-actin (left) and a CtsD immunoblot of wild-type and *Smpd1*⁻/⁻ d15.5 placentas (right). CtsD mature protein levels were unchanged between wild-type and *Smpd1*⁻/⁻ placentas. (E) Densiometric analysis of LC3-II to LC3-I (left) and a LC3 immunoblot of wild-type and *Smpd1*⁻/⁻ d15.5 placentas (right). The LC3-II:LC3-I ratio is significantly higher in *Smpd1*⁻/⁻ placentas ($P<0.05$). (F) Densiometric analysis of p62 protein expression normalized to β-actin (left) and a p62 immunoblot of wild-type and *Smpd1*⁻/⁻ d15.5 placentas (right). p62 is significantly increased in *Smpd1*⁻/⁻ placentas ($P<0.05$). Individual data points represent individual placentas obtained from at least two different dams; different letters represent statistical significance (unpaired *t*-test). Densiometric analysis was conducted using ImageJ. All data are mean±s.e.m. For details of the statistical test(s) used, see the Materials and Methods.

lysosomal enzyme, we first set out to determine how its deficiency affects the function and integrity of lysosomes in the placenta. Western blot analysis revealed a significant increase in the abundance of lysosomal marker, LAMP1 (Fig. 4A). In addition, immunofluorescence staining for LAMP1 confirmed drastic accumulation of enlarged lysosomes, especially in the $Smpd1^{-/-}$ labyrinth (Fig. 4B). To further analyze the lysosomal phenotype in $Smpd1^{-/-}$ placentas, we assessed the protein level of ATP6V1B2, a subunit of the V-ATPase that is responsible for acidification of lysosomes. Western blot analysis revealed reduced ATP6V1B2 expression in $Smpd1^{-/-}$ placentas (Fig. 4C). This suggests that lysosomal acidification, and therefore activity, may be impaired in $Smpd1^{-/-}$ placentas. Decreased catalytic processing of lysosomal proteases is another common sign of lysosomal dysfunction. We analyzed the processing of Cathepsin D (CtsD), a ubiquitous lysosomal protease, and found no difference in the levels of its fully mature form (Fig. 4D).

Autophagosome formation and subsequent fusion with lysosomes are key steps in autophagy. To investigate whether autophagosome formation and clearance is impaired in $Smpd1^{-/-}$ placentas, we analyzed the levels of the autophagosome marker LC3-II, as well as protein cargo receptor p62/SQSTM1. Levels of LC3-II (Fig. 4E) and p62 (Fig. 4F) were significantly increased in $Smpd1^{-/-}$ placentas, suggesting either increased autophagy or impaired autophagosome clearance. Thus, the enlarged lysosomes we observed within the labyrinth may be late autolysosomes that cannot be cleared from the cell due to accumulated sphingomyelin-blocking efflux. This is consistent with reports of $Smpd1$-deficient neurons (Gabande-Rodriguez et al., 2014), lungs and kidneys (Corcelle-Termeau et al., 2016) displaying problems with autophagy.

An accumulation of lysosomal and autophagosomal markers in $Smpd1^{-/-}$ placentas could be due to either increased biogenesis (enhanced autophagy), or defective clearance

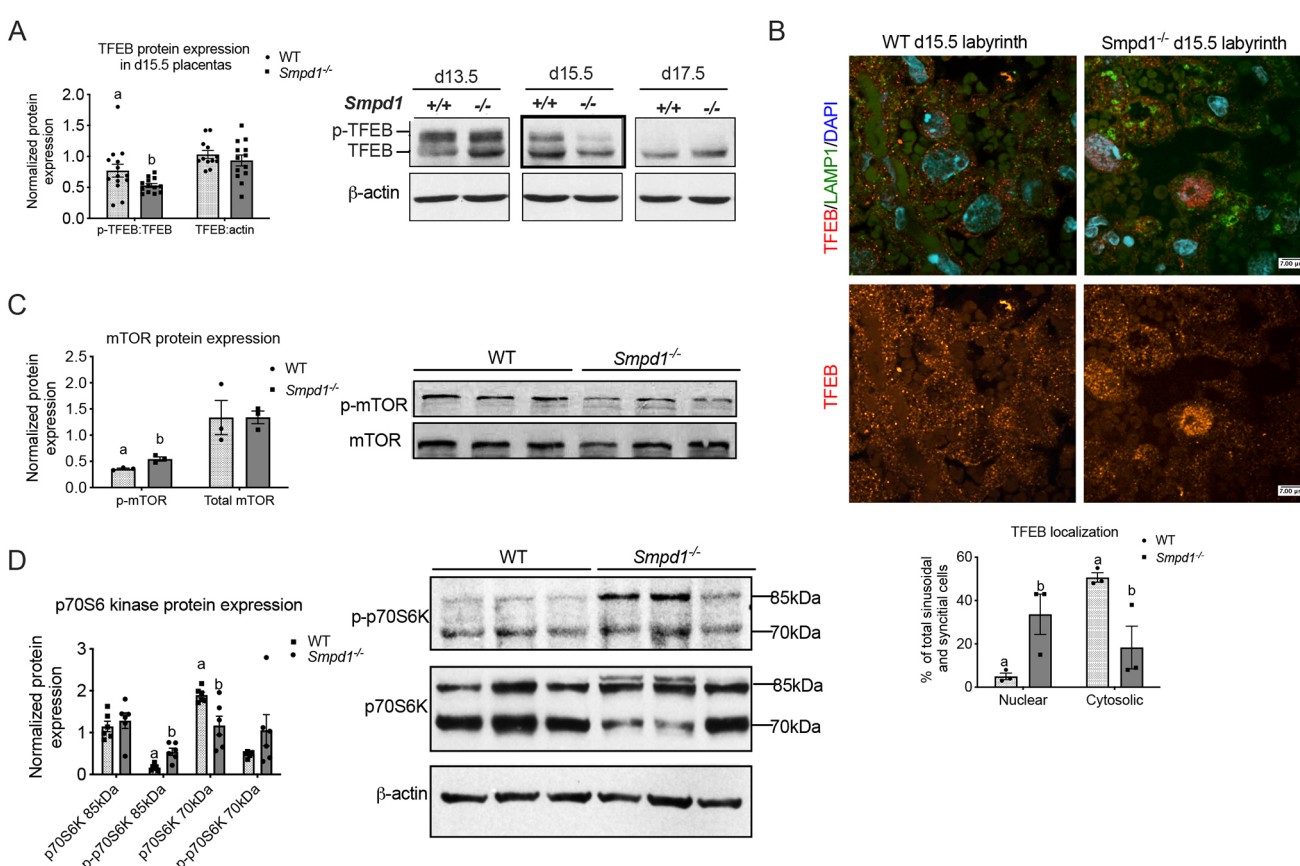

**Fig. 5. Increased autophagy in $Smpd1^{-/-}$ placentas.** (A) Densiometric analysis (left) of phospho-TFEB (p-TFEB) and total TFEB protein expression in gestational day (d) 15.5 wild-type and $Smpd1^{-/-}$ placentas, normalized to β-actin. P-TFEB protein levels are significantly decreased in $Smpd1^{-/-}$ placentas ($P<0.05$). Although slightly elevated in wild-type placenta, there was no significant change in total TFEB levels. (Right) A TFEB immunoblot of wild-type (+/+) and $Smpd1$-deficient (−/−) d13.5, d15.5 and d17.5 placentas. (B) (Top) Immunofluorescence analysis of TFEB localization examined by confocal microscopy in d15.5 wild-type and $Smpd1^{-/-}$ placentas at 40× magnification. TFEB (red) localizes to the cytoplasm of wild-type placental labyrinth and to the nucleus of labyrinth trophoblast in the $Smpd1^{-/-}$ placental labyrinth. Lysosomes immunostained by LAMP1 (green) and nuclei are stained with DAPI (blue). Scale bars: 5 μm. (Bottom) Quantitation of TFEB localization in sinusoidal and syncytial cells. $Smpd1^{-/-}$ placentas have a higher percentage of labyrinth trophoblasts, which have nuclear TFEB staining, compared to wild type ($P<0.05$). (C) A phospho-mTOR (Ser2448) and a mTOR immunoblot (right) of wild-type and $Smpd1^{-/-}$ d15.5 placentas, and densiometric analysis (left) of p-mTOR and mTOR protein expression normalized to β-actin. pmTOR protein level is significantly increased in $Smpd1^{-/-}$ placentas. (D) A Phospho-p70S6K (Thr389) and p70S6K immunoblot (right) and densiometric analysis (left). All data are mean±s.e.m. per placenta obtained from at least two different dams. Different letters represent statistical significance ($P<0.05$). For details of the statistical test(s) used, see the Materials and Methods.

(impaired autophagic flux). To determine whether lysosomal and autophagosomal production is enhanced, we analyzed the protein expression of transcription factor EB (TFEB), a transcription factor that positively regulates the expression of genes that encode proteins involved in lysosomal biogenesis and autophagy (Settembre et al., 2011). Under normal conditions, TFEB is retained at the lysosome or within the cytoplasm in its phosphorylated state, where the phosphorylation is maintained by several kinases, including mTOR (Settembre et al., 2012). Under conditions of nutritional deprivation or lysosomal stress, TFEB is dephosphorylated and translocates to the nucleus. Western blot analysis revealed that levels of phosphorylated TFEB were significantly reduced in *Smpd1*$^{-/-}$ placentas (Fig. 5A). This trend was stage dependent, and most obvious at d15.5, where almost no pTFEB was observed in the mutant placentas. In concurrence with our western blot findings, immunofluorescent staining revealed more nuclear localization of TFEB in *Smpd1*$^{-/-}$ labyrinth trophoblast, primarily among sinusoidal trophoblast giant cells (S-TGCs) (Fig. 5B).

Finally, protein levels of mTOR, the key upstream regulator of autophagy, as well as its downstream target p70S6 kinase, were

assessed. Levels of phosphorylated mTOR (Ser2448) were significantly increased in *Smpd1*$^{-/-}$ placentas (Fig. 5C). In addition, levels of the downstream mTORC1 target p70 S6 kinase were also elevated (Fig. 5D), with a decrease in the abundance of p70 isoform. Since serine 2448 of mTOR sits within a negative regulatory domain and is a target of p70S6K (Holz and Blenis, 2005), it is a part of autoregulatory loop that dampen mTOR kinase activity (Figueiredo et al., 2017). Decreased TFEB phosphorylation in this context is consistent with this negative feedback, indicating that decreased mTOR activity is a likely culprit responsible for inefficient lysosomal signaling.

## Intrauterine growth restriction fetuses cannot be rescued in a wild-type uterine environment

Phospholipids are secreted signaling molecules that can be delivered to placental tissues via maternal circulation. To establish whether the intrauterine growth restriction (IUGR) phenotype observed in *Smpd1*$^{-/-}$ fetuses is embryonic versus maternal in origin, we performed embryo transfer experiments. We placed wild-type and *Smpd1*$^{-/-}$ blastocysts into recipient females of the

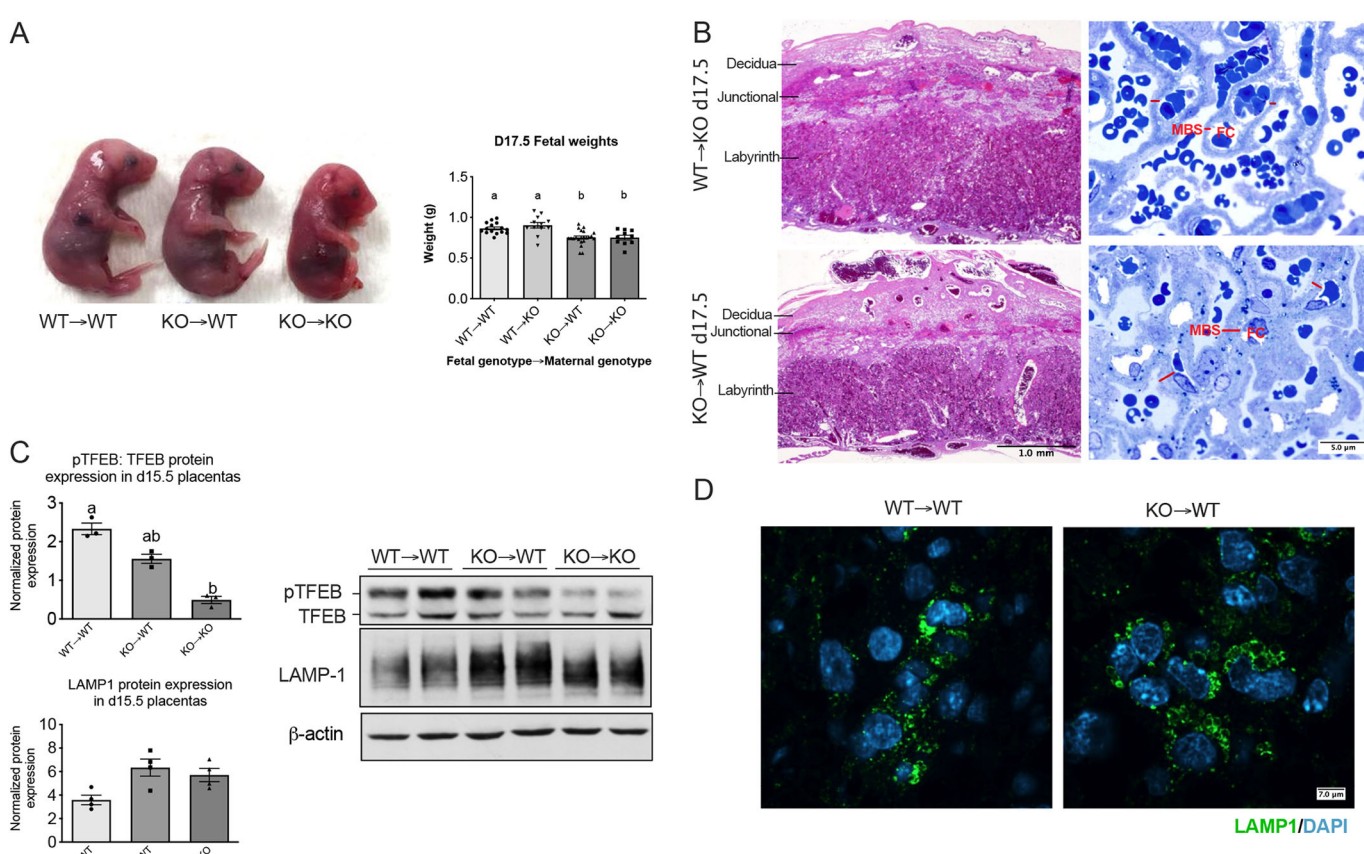

**Fig. 6. Fetal growth restriction of *Smpd1*$^{-/-}$ embryos cannot be rescued in a wild-type uterine environment.** (A) Reduced size of *Smpd1*$^{-/-}$ embryos grown in a wild-type uterus (KO→WT) at gestational day (d) 17.5, compared to wild-type and *Smpd1*$^{-/-}$ embryos grown in either a wild-type or *Smpd1*$^{-/-}$ uterus (WT→WT, WT→KO). *Smpd1*$^{-/-}$ (KO) embryos grown in a wild-type uterus (KO→WT) had significantly reduced weights at d17.5 compared with wild-type embryos grown in either a wild-type or *Smpd1*$^{-/-}$ uterus (WT→WT and WT→KO) ($P>0.01$). In addition, *Smpd1*$^{-/-}$ embryos grown in a wild-type uterus (KO→WT) had similar weights to embryos grown in a *Smpd1*$^{-/-}$ uterus (KO→KO). Individual weights were obtained from pups from at least three different dams (one-way ANOVA and Bonferroni post-hoc test). (B) d17.5 wild-type placenta grown in a *Smpd1*$^{-/-}$ uterus (WT→KO), stained with Hematoxylin and Eosin (2× magnification; top left) and d17.5 *Smpd1*$^{-/-}$ placenta grown in a wild-type uterus (KO→WT) stained with Hematoxylin and Eosin (2× magnification; bottom left). Toluidine Blue-stained semi-thin sections of d17.5 WT→KO labyrinth (top right) and d17.5 KO→WT labyrinth (bottom right) at 100× magnification. MBS, maternal blood space; FC, fetal capillary; red lines indicate interhemal thickness. (C) (Right) Protein expression of TFEB and LAMP1 in d15.5 WT→WT, KO→WT and KO→KO placentas. (Left) Densiometric analysis of TFEB and LAMP1 protein expression (Brown-Forsythe and Welch ANOVA test). (D) Representative LAMP1 immunofluorescence and localization of enlarged lysosomes in the placenta. Each value on the graphs represents data from an individual placenta, obtained from at least two different dams. Different letters represent statistical significance. All data are mean±s.e.m. For details of the statistical test(s) used, see the Materials and Methods. Scale bars: 1.0 mm in B (left); 5 μm in B (right); 7 μm in D.

opposite genotype. Weights of fetuses were measured at d17.5, since this is the gestational day where we observed the fetal restriction phenotype. Fetal weights of $Smpd1^{-/-}$ embryos remained significantly reduced, despite their development in a wild-type uterine environment (Fig. 6A). Labyrinthine pathologies of $Smpd1$-deficient embryos developed in a wild-type uterus (KO→WT) did not improve. Further inspection confirmed no obvious change in labyrinthine congestion, with a noticeably thicker interhemal membrane still present (Fig. 6B). Wild-type embryos developed in deficient dams appeared normal, and no obvious placental pathologies were observed. This is consistent with the growth restriction phenotype of $Smpd1^{-/-}$ fetuses being caused by placental defects and not by deficiency of maternally derived sphingolipids.

We also analyzed the key molecular changes associated with $Smpd1$ deficiency, such as lysosomal accumulation and TFEB phosphorylation, to determine whether these are predominantly embryonic, placental or maternal in origin. Levels of phosphorylated TFEB appeared partially corrected in KO placentas grown in wild-type uterus, while LAMP1 levels remained elevated (Fig. 6C). Immunofluorescent staining of placentas confirmed an accumulation of enlarged lysosomes (Fig. 6D). We thus conclude that the lysosomal dysfunction caused by SMpd1 deficiency is not triggered by a compromised maternal uterine environment, but is a result of trophoblast dysfunction.

## DISCUSSION

We show that acid sphingomyelinase ($Smdp1$) is required for proper placentation. Specifically, we observed decreased size of the labyrinth, which leads to a decreased area of exchange between maternal and fetal blood spaces, thus limiting the supply of nutrients to the fetus and resulting in IUGR. In addition, there are also structural labyrinthine changes, including thickening of the syncytial layers and reduction of maternal blood spaces. Moreover, we demonstrate that a deficiency in Smpd1 results in abnormal autophagy and impaired lysosomal function in labyrinthine trophoblast. Where these hallmarks are elevated as a consequence of poor nutrient uptake or a primary defect contributing to poor nutrient sensing remains to be established.

It is notable that the cell type with relatively high expression of Smpd1 – sinusoidal giant cells – are also most affected by altered TFEB localization and lysosomal swelling. It is possible that a subtle accumulation of sphingosine promotes lysosomal calcium leak, triggering a signaling cascade that ends with nuclear TFEB translocation (Hoglinger et al., 2015). S-TGCs are part of the interhemal membrane of the murine labyrinth, lining the maternal blood spaces. These cells have a distinct developmental origin among TGC subtypes and are well situated to facilitate the delivery of factors into the maternal circulation. Interestingly, genetic ablation of sinusoidal TGCs results in fetal growth restriction and embryonic lethality by d18.5 (Outhwaite et al., 2015). Placentas lacking S-TGCs exhibit significant hemorrhaging and degradation, indicating that these cells play an important role in maintaining the vasculature within the labyrinth. SMPD1 is one of the enzymes involved in the production of S1P – a sphingolipid important for vascular development – it is possible that S-TGCs allow the proper production of S1P and thus maintain the function of the fetal/maternal vasculature.

Several studies have reported on circulating levels of S1P during human pregnancies with variable outcomes of either an increase (Melland-Smith et al., 2015) or no change (Johnstone et al., 2023). While we have not analyzed circulating S1P, we detected a decrease in placental S1P in tissue lacking $Smpd1$. Circulating and tissue levels are not the same, as different cells types contribute to the S1P pool. Consistent with our findings, recent reports detected increased levels of sphingomyelin and trend towards decreased levels of S1P in placental chorionic arteries isolated from pre-eclamptic placentas. (Del Gaudio et al., 2020).

Lysosomal dysfunction could also be a contributing factor to the nutrient deprivation state of the labyrinth. We found no changes in the processing of ubiquitous cathepsin (cathepsin D) in $Smpd1$-deficient placental lysates. However, the lysosome contains over 50 hydrolytic enzymes and the degree of lysosomal acidification or dysfunction may affect certain enzymes more so than others. This is consistent with lysosomal damage in $Smpd1^{-/-}$ neurons, resulting in excess cytosolic release of cathepsin B (CtsB), while no cytosolic release of Cathepsin D was detected (Gabande-Rodriguez et al., 2014). Therefore, alterations in the processing and activity of different lysosomal proteases may be tissue or cell-type specific.

Consistent with our observations, attenuated mTOR signaling has been reported to result in increased syncytialization with concomitant poor fetal growth both in mouse and human (Shao et al., 2021). In addition, TFEB has recently been reported to regulate syncytialization and directly transactivate other targets beside lysosomal genes, including SynA (Cesana et al., 2024; Zheng et al., 2024). Previous studies have shown that null $SynA$ embryos are severely growth restricted and die $in$ $utero$ (Dupressoir et al., 2009). Deletion of SynA or its receptor Ly6 (Langford et al., 2018) produced a thickened interhemal membrane with defective formation of ST-I, which subsequently impairs the formation of ST-II. Interestingly, a placenta with disruption of a second fusogenic driver, $Synb,$ exhibits impaired formation of ST-II, with fetuses displaying only a mild growth impairment (Dupressoir et al., 2011). Thus, decreased expression of $Syna$, potentially due to defective TFEB signaling in $Smpd1$ placentas, may be the contributing factor causing impaired syncytial fusion and thus thickened interhemal distance in $Smpd1$ mutants.

Together, these findings demonstrate that numerous placental defects drive delayed fetal growth. While phosphorylation of mTOR/TFEB is partially sensitive to maternal environment, the accumulation of lysosomes and defects in syncytial differentiation in $Smpd1^{-/-}$ placentas are an intrinsic cellular defect that cannot be rescued by uterine environment. The limitation of our whole animal deletion model is that the whole conceptus is $Smpd1$ deficient. We thus cannot exclude the possibility that placenta is not the only contributing factor to poor fetal growth. In addition, placental histological findings should be re-examined in a sex-specific manner, as we observed preferential growth restriction in surviving female pups. In any case, Smpd1-deficient mice can be used as a model of IUGR to further our understanding of molecular pathways and placental pathologies associated with abnormal fetal growth.

## MATERIALS AND METHODS
### Mice
The mice used in these experiments were maintained on a 12 h:12 h light-dark cycle with free access to food and water. Mouse experiments were performed in accordance with the Canadian Council on Animal Care (CCAC) guidelines for the Use of Animals in Research and Laboratory Animal Care, under protocols approved by the animal care committees of The Centre for Phenogenomics (TCP).

Wild-type and mutant mice (officially known as $129.Smpd1^{tm1Esc}$ and referred to as $Smpd1^{-/-}$ or KO in this study) of mixed C57Bl/6 and 129/Sv background were obtained from the a breeding colony of Dr Schuchman (Horinouchi et al., 1995). $Smpd1$ FLAG-tagged mice ($C57BL/6N$-$Smpd1^{em11Tcp}$, referred to as $Smpd1$-FLAG in this study) were generated

at the TCP using CRISPR/Cas9 and registered at MGI with allele number 7565732. A N-terminal 3×FLAG tag followed by a flexible linker (GGGGS) was inserted adjacent to the ATG codon of Smpd1. For collection of wild-type (Smpd1[+/+]) and KO (Smpd1[−/−]) placentas, females of ~6-8 weeks of age were mated with males of the identical genotype. Gestational age was determined based on the presence of a vaginal plug, with the morning of detection designated as gestational day 0.5 (d0.5). Pregnant dams were euthanized at d11.5, d13.5, d15.5 and d17.5, and the number of conceptuses and resorptions were recorded. Fetuses and placentas were removed from the uterus and their weights were recorded. For reciprocal embryo transfer experiments, recipient dams were hormonally primed with 2.5 IU of PMSG/hCG (Folligon/Chorulon; Intervet/Merck, Canada) and mated to vasectomized males. Donor females were primed with 5 IU of PMSG/hCG and mated to a male of the same genotype. Blastocysts were collected at d3.5 and immediately transferred into uterus of pseudo pregnant recipient females.

### Immunohistochemistry
Slides were deparaffinized in xylene and rehydrated in a descending alcohol gradient. Antigen retrieval was performed using 10 mM Tris-Base and 1 mM EDTA buffer (pH 8.0) at 100°C for 15 min. After blocking, sections were then incubated overnight at 4°C with rabbit anti-DYKDDDDK antibody (Rockland, 600-401-383; 1:200) diluted with 5% horse serum in PBS. The following day, slides were washed followed by incubation with biotinylated anti-rabbit secondary antibody, detected using SignalStain DAB Substrate Kit (13079, Cell Signaling) and counterstained with Hematoxylin.

### Terminal deoxynucleotidyl transferase dUTP nick-end labeling
TUNEL staining was performed as previously described (Detmar et al., 2019). Briefly, d15.5 and d17.5 wild-type and Smpd1[−/−] placentas were collected, fixed in 10% formalin. All tissue blocks were serially sectioned at 5 μm, and sections corresponding to the midline, and 50 μm and 100 μm from the center were chosen for TUNEL staining. A light microscope (BX61, Olympus) was used for analysis, and the genotype and gestational time point were not made known to the researcher. All TUNEL-positive TGCs were counted in the entire conceptus.

### Immunofluorescence
Slides were processed as outlined above and citrate buffer (pH 6) antigen retrieval was performed in a steamer. Subsequently, slides were quenched in a solution of 0.1% Sudan Black in 70% ethanol for 15 min, and then washed in PBS. Sections were then blocked with 10% horse serum in PBS for 1 h at room temperature and incubated overnight at 4°C with primary antibodies diluted with 5% horse serum in PBS. The following day, slides were washed and incubated with a host-specific secondary antibody conjugated with Alexa Flour dyes (Invitrogen Life Technologies) and counterstained with blue fluorescent 4′,6-diamidino-2-phenylindole (DAPI, Sigma). To analyze syncytiotrophoblast morphology, sections were co-stained with syncytiotrophoblast-I marker Mct1 (1:500, AB1286I, Abcam) and syncytiotrophoblast-II marker Mct4 (1:500, AB3314P, Abcam). Lysosomes were assessed by staining for lysosomal membrane marker LAMP-1 (1:200, ID4B, DSHB) and TFEB (1:200, A303-673A, Bethyl Laboratories). The secondary antibodies used were donkey anti-rabbit IgG (Alexa Fluor 594, Molecular Probes), donkey anti-chicken (488, Jackson Immunologicals) and donkey anti-rat IgG (DyLight 488; Bethyl).

### Semi-thin sections and interhaemal distance
Pregnant wild-type and KO dams were euthanized at d17.5 and four placentas from each dam were collected as described above. An ~1 mm³ piece of labyrinth adjacent to the umbilical cord was dissected from each placenta, as well as an ~1 mm³ piece of cortex from each kidney. Placental pieces were fixed separately in 2% gluteraldehyde overnight, then dehydrated and embedded in plastic resins. Embedded labyrinth tissues were sectioned at 1 μm thickness (semi-thin sections) and stained with Toluidine Blue for stereological analysis. Sections of 50 nm thickness (thin sections) were viewed using a transmission electron microscope (Model CM100; FEI). Interhemal membrane thickness was measured using the

orthogonal intercept method on the NewCAST program from Visiopharm (Hoersholm, Denmark). Semi-thin labyrinth tissue sections were obtained as described above and viewed at 60× magnification on a light microscope (BX61, Olympus KeyMed). The NewCAST program superimposes a grid of equidistant parallel lines on the image. When a line intersects the trace of a maternal blood space, the shortest linear distance to a fetal capillary is measured and expressed as hormonic mean. A total of 200 measurements from sections of each genotype were made.

### Stereology and vascular casts
Serial sections of d15.5 wild-type and KO placentas were stained with Hematoxylin and Eosin, and viewed on a light microscope (BX61, Olympus KeyMed) at 10× magnification. The areas of individual placenta regions (decidua, junctional zone and labyrinth) were measured using the Cavalieri principle, which is a point-counting method that superimposes a grid of cross hairs over an image, with each cross hair representing a defined area (mm²). The cross hairs were labelled either decidua (D), junctional zone (J) or labyrinth (L) if they fell on the respective region of the placenta. The total number of labels per region was determined and multiplied by the defined area of each cross hair to calculate the total area (mm²) of each individual layer. Total area counts were averaged per placenta among the three midline serial sections spaced 100 μM apart. Hematoxylin and Eosin-stained slides were scanned and analyzed using QuPath-0.50-x64. For glycogen cell counts, entire placental sections were used; for maternal space quantitation, six randomized rectangular regions were selected for each placenta. Final values were determined through the ratio of counts to total area.

Vascular Cast analyses were performed on specimens obtained from fetuses collected from three wild-type and five KO dams at day 15.5 of pregnancy. A detailed description of the method for the injection of the radiopaque silicone rubber X-ray contrast agent (Microfil, Flow Tech, Carver, MA) into the fetoplacental vasculature as well as micro-CT imaging have been previously described by Rennie et al. (2011, 2012). Vascular surface renderings were generated from micro-CT data to visualize the arterial vasculature, and vascular segmentation was performed as outlined by Rennie et al. (2007).

### Western blotting
Placental protein lysates were prepared from tissues collected from at least two different dams of each genotype. Individual placentas containing decidua were mechanically disrupted in 1% SDS-RIPA buffer containing complete protease inhibitor cocktail (Sigma-Aldrich) and 1× phosphatase inhibitor cocktail (Sigma-Aldrich). Protein concentrations were determined using the BCA protein assay. Protein samples were resolved through standard denaturing acrylamide gels of various concentrations, prepared according the manufacturer's instructions (BioRad). Proteins were transferred to a BioTrace nitrocellulose membrane, blocked for 1 h at room temperature with 5% skim milk in PBS with 0.1% Tween-20 (PBST), then incubated at 4°C with various primary antibodies. Following primary antibody incubation, membranes were washed in 0.1% PBS-T, then incubated with species-appropriate horseradish peroxidase (HRP)-conjugated secondary antibodies. Membranes were placed in enhanced chemiluminescence HRP substrate (Western Lighting Plus ECL, PerkinElmer), exposed and analyzed using ImageJ software. Primary antibodies used for western blotting were anti-β-actin, goat polyclonal (I-19) (Santa Cruz Biotechnology,sc-1616; 1:500), anti-LAMP-1 (DSHB, 1D4B; 1:1000), anti-TFEB (Bethyl, A303-673A; 1:2000), anti-mTOR Ser2448 (Cell Signaling, 2971; 1:800), anti-mTOR (Cell Signaling, 2972; 1:1000), anti-p62 (PROGEN Biotechnik, GP62-C; 1:1000), anti-LC3 (MBL, PM036; 1:1000), anti-ATP6V1B2 (Proteintech, 15097-1-AP; 1:1000), anti-Cathepsin D (Bio-Vision, 3191R-100; 1:2000), anti-DDDK-HRP (Proteintech, HRP-66008; 1:15,000), anti-p70 S6 kinase Thr389 (Cell Signaling, 9205; 1:800) and anti-p70 S6 kinase (Cell Signaling, 9202; 1:1000). Secondary antibodies were anti-goat HRP (AbD Serotec, 642001; 1:5000), anti-rabbit HRP (Cell Signaling, 7074; 1:5000) and anti-mouse HRP (Bio-Rad, 1706516; dilution 1:10,000).

### Lipidomic profiles
Wild-type and Smpd1 KO placentas with decidua were collected at d15.5 and stored at −80°C. Whole placentas (three wild type and three KO) each

obtained from three different dams were used and analyzed in two batches. Lipid extraction (performed according to Bligh and Dyer, 1959), mass spectrometry and lipidomic analysis were conducted by the Analytical Facility for Bioactive Molecules (AFBM) at the Hospital for Sick Children, Toronto, ON, Canada. Details of the extraction and analytical procedure have been described previously by Melland-Smith et al. (2015). Results were batch corrected and are shown as fold change of wild-type placentas.

### Smpd1 and sphingosine kinase activity assays
Smpd1 enzymatic activity was measured using the Echelon Acid Sphingomyelinase Assay Kit (K-3200, Echelon Bioscience). The protocol we developed has been submitted to the manufacturer and is included in their instruction manual [TDS_K-3200_Rev10 (echelon-inc.com)]. Briefly, placentas at various gestational time points were collected and stored immediately at −80°C. Individual placenta layers (labyrinth, junctional zone and decidua) were also collected at d15.5. Samples were homogenized in substrate buffer (K-3203) and a BCA protein assay was performed to determine the total protein concentration per sample. Approximately 7 µg of total protein from each sample was used in the assay. The reaction was stopped after incubation with the A-SMase substrate at 37°C for 3 h and analyzed using a fluorescence plate reader (Tecan Infinite M200). The specificity of the kit was tested by running KO ($Smpd1^{-/-}$) brain tissue, which produced no detectable signal.

Sphingosine kinase (SPHK) enzyme activity was measured using the Echelon Sphingosine Kinase Activity Assay Kit (K-3500, Echelon Bioscience) according to manufacturer's instructions. To test the specificity of the kit, samples were incubated with a SPHK1 inhibitor (SKI-II) (10009222, Cayman Chemical), which produced no detectable signal.

### Quantitative real-time PCR
RNA was extracted from placentas using the TRIzol method, according to the manufacturer's instructions. The extracted RNA was treated with DNase-I (Sigma-Aldrich) to remove DNA contaminants and then reverse transcribed into cDNA using a reverse transcription kit (Thermo Fisher, 4368814) following the manufacturer's protocol. All gene expression experiments utilized the Wisent Advanced qPCR master mix and were run using CFX96 Real-Time System (Bio-Rad C1000 Thermal Cycler). Table S1 below provides a summary of the primer sequences, as well as the respective annealing temperatures used for each target gene. For analyzing *Smpd1* expression in wild-type placentas across gestation, expression levels were normalized to the geometric mean of three housekeeping genes (*Actb*, *Hprt1* and *Tbp*). For comparisons between wild-type and KO placentas, expression levels were normalized to two housekeeping genes (*Actb* and *Pgk1*) as *Hprt1* and *Tbp* proved unstable in *Smpd1*-deficient tissues.

### Statistical analysis
Statistical analysis was performed using GraphPad Prism Software. Data groups were first tested for normality, using the Shapiro-Wilk test. If the data were distributed normally, statistical analysis was performed using either an unpaired *t*-test for comparing two groups (with Welch's correction if groups had unequal variance), and a one- or two-way ANOVA followed by either a Bonferroni post-hoc test for comparison or Holm Sidak comparison of multiple groups. In the cases where datasets were not normally distributed, a Mann–Whitney rank sum test was used to compare datasets with two groups, and the Kruskal–Wallis test followed by a Dunn's multiple comparison test was used for comparison of multiple groups. GraphPad Prism 10.3.1 was used for normality and statistical analyses. All data passed the Shapiro-Wilk normality test, and an unpaired *t*-test was performed. *$P \leq 0.05$, **$P \leq 0.01$. Data are mean±s.e.m., with $P<0.05$ being defined as significant.

### Acknowledgements
We thank Drs Helen McNeil and Brian Cox, and both past and current members of Jurisicova laboratory for helpful discussions and feedback during the course of preparation of this manuscript. We also thank Sonia Encinas and Maryam Pashei for their assistance with tissue collection during revision of this manuscript, and Dr Ying Zhang and Qing Wu for histological assessment of collected tissue. Additional thanks goes to Marina Gertsenstein, Dr Lauryl Nutter and members of The Centre for Phenogenomics for the generation of Smpd1-Flag mice and guiding our embryo transfer experiments.

### Competing interests
The authors declare no competing or financial interests.

### Author contributions
Conceptualization: A.J., J.S.; Data curation: A.J., I.R., K.S., H.L.; Formal analysis: I.R., K.S., R.M., S.S., N.T.; Funding acquisition: A.J.; Investigation: A.J., H.L., R.M., S.S., Q.W.; Methodology: A.J., N.T.; Software: J.S.; Supervision: J.S.; Validation: R.M., Q.W.; Writing – original draft: I.R.; Writing – review & editing: A.J.

### Funding
The research behind this article was supported by funding from the Natural Sciences and Engineering Research Council of Canada (RGPIN-04497) and the Canadian Institutes of Health Research (FRN 156081) awarded to A.J. In addition, I.R., K.S. and S.S. were partially supported by the Lunenfeld Tanenbaum Research Institute-Ontario Student Opportunity Trust Fund (LTRI-OSOTF) program. Open Access funding provided by the University of Toronto. Deposited in PMC for immediate release.

### Data and resource availability
All relevant data and details of resources can be found within the article and its supplementary information.

### The people behind the papers
This article has an associated 'The people behind the papers' interview with some of the authors.

### Peer review history
The peer review history is available online at https://journals.biologists.com/dev/lookup/doi/10.1242/dev.204425.reviewer-comments.pdf

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
