## [Peer Review File · Development (Cambridge, England)]

Acid sphingomyelinase is a gatekeeper of placental labyrinthine architecture and function.

Isidora Rovic, Katherine Szelag, Han Li, Rosanne McQuaid, Sara Sugin, Que Wu, Natalia Theodora, John Sled and Andrea Jurisicova
DOI: 10.1242/dev.204425

Editor: Anna-Katerina Hadjantonakis

Review timeline

Original submission:	30 September 2024
Editorial decision:	7 November 2024
First revision received:	1 May 2025
Editorial decision:	20 May 2025
Second revision received:	29 July 2025
Accepted:	12 August 2025

Original submission

First decision letter

MS ID#: dev.204425

MS TITLE: Acid sphingomyelinase is a gatekeeper of labyrinthine architecture and function.

AUTHORS: Andrea Jurisicova; Isidora Rovic; Katherine Szelag; Han Li

Dear Dr Jurisicova,

I have now received all the referees' reports on the above manuscript, and have reached a decision. The referees' comments are appended below, or you can access them online: please go to:

As you will see, the referees express considerable interest in your work, but have some significant criticisms and recommend a substantial revision of your manuscript before we can consider publication. If you are able to revise the manuscript along the lines suggested, which may involve further experiments, I will be happy receive a revised version of the manuscript. Your revised paper will be re-reviewed by one or more of the original referees, and acceptance of your manuscript will depend on your addressing satisfactorily the reviewers' major concerns. Please also note that Development will normally permit only one round of major revision. If it would be helpful, you are welcome to contact us to discuss your revision in greater detail. Please send us a point-by-point response indicating your plans for addressing the referees' comments, and we will look over this and provide further guidance.

Please attend to all of the reviewers' comments and ensure that you clearly highlight all changes made in the revised manuscript. Please avoid using 'Tracked changes' in Word files as these are lost in PDF conversion. I should be grateful if you would also provide a point-by-point response detailing how you have dealt with the points raised by the reviewers in the 'Response to Reviewers' box. If you do not agree with any of their criticisms or suggestions please explain clearly why this is so.

Reviewer 1

Advance summary and potential significance to field

In the current manuscript by Rovic et al., the authors investigate the effects of knocking out the acid sphingomyelinase (Smpd1) gene and describe a placental phenotype that results in mild IUGR late in gestation. The authors also perform mass spec analysis to identify the lipid changes as a result of SMPD1 ablation. The authors find reduced ceramide levels, as expected from SMPD1's function in hydrolyzing sphingomyelin into ceramide. Surprisingly, the authors observe decreased levels of sphingosine-1-phosphate (S1P), a sphingolipid important for promoting angiogenesis. They also observe several hallmarks of defective autophagy and lysosomal impairment in Smpd1^{-/-} placentas. Finally, the authors tie this phenotype at least in part to reduced phosphorylation levels of the transcription factor TFEB that, in its unphosphorylated form, enters the nucleus and drives expression of genes involved in lysosomal biogenesis and autophagy.

Overall, this is an interesting study that provides a substantial amount of molecular detail to explain the phenotypes observed. The relevance of sphingomyelin-related lipids exhibiting dysregulation in human IUGR placentas strengthens the relevance of this study.

There are, however, a number of issues that should be addressed:

The anti-FLAG IHC staining is problematic as it is very brown everywhere. Also, in the supplement, the junctional zone staining is only prominent on one side of the placenta. Section quality as well as staining quality should be improved. Also, please show a junctional zone magnification in the main figure, as this is specifically mentioned in the text.

It is not entirely clear why the authors have this unspecific band, as the FLAG antibody is routinely used for clean Western blotting. It should be attempted to improve on the WB.

Figure 1a-c consistently shows a peak of Smpd1 expression at E13.5, yet the authors state that no changes were observed over the course of pregnancy. Please clarify this statement.

Do the authors see a sex bias in the pups and/or offspring? Please add this aspect so to ascertain (or discount) whether the increased embryonic loss is preferentially affecting male or female pups.

Figure 3, the MCT1/MCT4 figure element is partially obscured by the EM image above. The MCT1/MCT4 stainings are of poor quality. The authors should provide better quality staining with stronger signal intensities, so to help the reader see the phenotype.

In general, the figure assembly is fairly unappealing with large bar graphs but very small embryo photos, poor alignment between blots and quantifications, and the like. The authors should spend some time to improve on the displays.

Finally, along similar lines, the manuscript needs to be edited for text errors throughout, including the figure legends.

Reviewer 2*Advance summary and potential significance to field*

The primary objective of this study is to investigate whether sphingomyelin-phosphodiesterase 1 (Smpd1) is required for normal placental development in the mouse. The authors found that deficiency of Smpd1 results in intrauterine growth restriction (IUGR) was associated with reduced placental labyrinthine compartments and increased fetal-maternal interhaemal distance. Sphingosine 1-phosphate (S1P), produced downstream of Smpd1, is reduced three-fold. The placentas from Smpd1 deficient mice also show defective autophagy and lysosomal impairment.

This is an interesting novel study that evaluates the role of an enzyme that is upstream of two important signaling components of the cells with opposing effects. S1P is a cell survival factor while ceramide is pro-apoptotic. The results of the study support the authors' conclusions and the study is well done.

The few papers that have examined the role of SMDP1 in the placenta have mostly been in those from human beings with the inherent limitations. This study advances the field by using a mouse model where Smpd1 has been knocked out to determine the impact on placental development at various gestational ages. It is the first to find stable expression and activity of Smpd1 throughout gestation. This was facilitated by the authors' development of a Flag-tagged Smpd1 knock in mouse. The investigations of the mouse placenta structures is thorough and the study of autophagy and abnormal lysosomal morphology strongly supports the previous study done in human placentas. The embryo transfer experiments provide additional support for placental defects as the cause for IUGR in the Smpd1^{-/-} fetuses.

Comments for the author

1. A suggestion is to have the English carefully checked. There are numerous instances of missing articles (e.g. "the") within sentences and a few instances of awkward or incorrect wording.
2. Please define TFEB within the abstract.
3. A suggestion is to add the word, "placental" to the title to define the labyrinthine compartment.
4. Please add a paragraph to describe the placenta and its compartments etc. since this study is focused on "normal placental development" and the role of Smpd1. So for example, define the role of the compartments within the mouse placenta and also the importance of the increased thickness of the fetal-maternal interhaemal distance. This is not specifically described and for those who are not placental researchers, this would be important information.
5. In the figure legends, there is an indication that the numbers in the graphs represent the number of placentas (Figs 1, 3, 4, 5, 6, 7). Please define this, i.e. Would all of those placentas come from the same dam? Or would these be from separate dams? The 'n' value should be the dams, not the placentas.
6. What does the red arrow in Figure 1e represent? Please add this to the figure legend.
7. Figure 2 legend, "Smpd1^{-/-} fetuses exhibit reduced birth weights" is not a correct statement. Fetuses cannot have "birth" weights. Please reword.
8. Figure 3 - please indicate the Spg and Lab and Dec areas on the images in 3a.
9. The statement on Page 9, beginning with "Given that Smpd1 regulates the production of S1P..." does not take into account the many other enzymes involved in both production and degradation of S1P downstream of Smpd1 activity. Please revise this statement to reflect this.
10. Please expand on the Lipidomic Profile methods used and provide a relevant reference. More details are needed for the method. How many dams? How many placentas per dams? What is meant by fold change compared to WT? How was this done? Were all WT placentas averaged and this used to do the fold change?

Reviewer 3

Advance summary and potential significance to field

The authors uncover novel functions for Smpd1, a key enzyme of the sphingolipid metabolic pathway, during placental development in mice. Smpd1 LOF results in decrease size of the fetal labyrinth, increased inter-hemal thickness, and IUGR at E17.5. The authors also delineate specific defects in the sphingolipid pathway and show that in their embryo transfer experiments that the IUGR is due to LOF in the embryo/placenta and not due to maternal LOF. There are a number of concerns with the completeness of this study, and with the Smpd1 flag/flag mouse model used, that are outlined below in the Major Comments section.

Comments for the author

Major comments

1. It is unclear if the *Smpd1* flag/flag mouse is an appropriate model to delineate the spatial expression pattern of SMPD1 in the placenta, considering that a prominent non-specific band showed on the Western blot. The low magnification image in Suppl. Fig. 1 does not allow to determine if there is also signal in the WT control. Higher magnification images would be required to exclude non-specific signals. Fig. 1e should include WT controls. A quick search indicated that polyclonal anti-SMPD1 antibodies are available and suitable for IHC. In situ hybridization would be another option.
2. While the study focuses on cells of the trophoblast lineage in the fetal labyrinth, the potential contribution of the junctional zone (JZ) trophoblast is not investigated. This is important since the JZ size is increased in the LOF mice. Note that the JZ contains both, spongiotrophoblast and glycogen trophoblast, the latter of them contain glycogen storage vesicles. It will be important to determine if SpT, GlyT or both cell types are more abundant in the mutant JZ.
3. It is unclear if *Smpd1* is expressed in fetal endothelial cells of the labyrinth, and whether its LOF results in vascular phenotypes in the placenta. This is particularly relevant, since the authors state that S1P acts downstream of *Smpd1* in the sphingolipid signaling pathway and is important for promoting angiogenesis. To address this, the authors should perform co-staining with an endothelial marker, e.g. CD31/PECAM.
4. In addition to the point #3, CD31 / PECAM staining will allow to determine the size of maternal and fetal blood space in the LOF placentas. Figure 7b suggests that both, maternal and fetal blood spaces are reduced, but this should be quantified.
5. Please clarify if WT controls shown in Fig. 2 are in the same mixed genetic background as the LOF mutant mice? Otherwise, the differences shown here could be due to strain-specific effects on the fetal weights.
6. The authors show in Figure 4 decreased Sphingosine kinase activity and decrease S1P activity in mutant placentas. The authors should discuss the discrepancy between human studies correlating high circulating S1P levels with pregnancy disorders including IUGR and their mouse model with reduced S1P levels and the IUGR phenotype.
7. Fig. 7: without staining for SpT or GyT specific markers, the authors cannot conclude that the JZ invaginations shown in panel 7b are SpT.
8. In their discussion of the results, and also in the abstract, the authors should state the hypothesis that the decreased size of the labyrinth leads to a decreased area of exchange between maternal and fetal blood space, thus limiting supply of nutrients to the fetus and resulting in IUGR.

Minor comments

1. All Figures: add error bars to the graphs
2. Fig. 2a: label images of embryos in the box next to the graph.
3. Fig. 3: labelling of panels does not align with the panels.
4. State in M&M if placental tissue used for Westerns, lipidomics profiles, and qRT-PCR contain decidua of just the fetal part of the placenta.
5. Fig.4: provide in the figure legend the full names of the enzymes shown in Fig. 4c.
6. Authors should replace "interhemal membrane" in the text with "interhemal distance" as used in the abstract.

First revision

Author response to reviewers' comments

We would like to thank the reviewers for critical evaluations and feedback how to improve our manuscript. We have performed new experiments and attempted to answer as many questions as possible with the tools that are available.

Reviewer 1:

1, The anti-FLAG IHC staining is problematic as it is very brown everywhere. Also, in the supplement, the junctional zone staining is only prominent on one side of the placenta. Section quality as well as staining quality should be improved. Also, please show a junctional zone magnification in the main figure, as this is specifically mentioned in the text.

We have tried (and continue to do so) to improve the quality of immunostaining. However, this is very challenging as Ab specificity is problematic. Original work that generated these images was performed with polyclonal anti DYKDDDDK rabbit Ab (Rockland, 600-401-383) several years ago. From our archived images, we choose these two examples (Flag on the top) along with images of wildtype (bottom) gestational age and area matched tissues stained at the same time (see below). We have also purchased this Rockland Ab however; the current lot is not performing as the last one. We did include another image of WT and Flag/flag placenta from day 13.5 in Suppl 1A. from low mag

Similarly, immunoreactivity was not specific, when anti Smpd1 antibody (Proteintech 14609-1-AP, or Novus NBP2-20426) were tested on WT and KO tissues using paraffin embedded or cryo sections.

2. It is not entirely clear why the authors have this nonspecific band, as the FLAG antibody is routinely used for clean Western blotting. It should be attempted to improve on the WB.

We have tested numerous FLAG /DDDDK Antibodies (clone M2, F1804 Sigma; Cell signaling 2368; Abcam ab1257; GeneScript A00170, GeneTex GTX115043). All these Ab produced non-specific signal on both immunostaining as well as on Western blots (WB). Non-specific nature of FLAG binding is also evidenced by well-know collection of non- specific targets that are identified in AP-MS studies (CRAPome <https://www.nature.com/articles/nmeth.2557>). We would like to point out that there are only handful of studies where FLAG tagging was used in live animals. In addition, majority of published papers referred to by the reviewer are overexpression studies, where targeted protein is many times upregulated. Thus, very short exposure is required to obtain the tagged signal. This is the reason why non-specific bands do not show up on exposed WB. We have used knock-in approach, where FLAG tagging is used to reflect endogenous level of expression, which unfortunately comes at cost of longer exposure and thus detection of nonspecific bands.

The best performing DDDDK (FLAG) Ab in our hands at the present time is Proteintech (HRP-66008) conjugated antibody (clone 8H6A10), which we have used to quantitate the protein levels using western blot and generate the new figure for Smpd1 protein abundance during development. Images show WB for WT and Flag/Flag placental lysates at day 17.5. ~75 kDa band is the anticipated mol weight of tagged version of Smpd1. Lower bands in WT tissues are not specific.

3. *Figure 1a-c consistently shows a peak of Smpd1 expression at E13.5, yet the authors state that no changes were observed over the course of pregnancy. Please clarify this statement.*

We re-did Western blot data where we used the above-mentioned anti-Flag Ab antibody on a larger number of tissues, and we observed increase in Flag reactivity at day 13.5, followed by a gradual decline through day 15.5 and 17.5. We have modified the statement in the manuscript accordingly, and we will include images of WB using antibody shown above.

Do the authors see a sex bias in the pups and/or offspring? Please add this aspect so to ascertain (or discount) whether the increased embryonic loss is preferentially affecting male or female pups.

This is a good question. Unfortunately, small size of our animal colony prevented us from performing large scale fetal studies. KO animals only breed and carry pregnancies for a short period of time (between 6-10 weeks of age), as they start showing signs of disease (tremors and locomotive changes) around 12 weeks of age. We do not see distorted ration of sexes at birth.

However, we did analyse offspring born from WTxWT and KOxKO pregnancies at day 2 postnatally. We observed that decreased weight was obvious for ASM KO females; there is significant interaction between genotype and sex ($p=0.039$; 2- Way ANOVA). These data are included as a supplemental figure (S.1) N= 4 WT moms (n=23 males and

18 females born in 7 litters). KO data were obtained from 2 dams (7 males and 7 females from 3 litters).

Reviewer2.

Thank you for the suggestions. We have made changes as you indicated in your comments.

1. A suggestion is to have the English carefully checked. There are numerous instances of missing articles (e.g. "the") within sentences and a few instances of awkward or incorrect wording.

2. Please define TFEB within the abstract.

Thank you; we have added the definition as suggested.

3. *A suggestion is to add the word, "placental" to the title to define the labyrinthine compartment.*

Added, as suggested.

4. Please add a paragraph to describe the placenta and its compartments etc. since this study is focused on "normal placental development" and the role of Smpd1. So for example, define the role of the compartments within the mouse placenta and also the importance of the increased thickness of the fetal-maternal interhaemal distance. This is not specifically described and for those who are not placental researchers, this would be important information.

Paragraph added: Mouse placenta is organized into 3 distinct regions - labyrinthine compartment, a site of fetal/maternal nutrient exchange, junctional zone enriched in spongiotrophoblast and glycogen cells important for hormonal and metabolic function and maternally derived decidua.

5. In the figure legends, there is an indication that the numbers in the graphs represent the number of placentas (Figs 1, 3, 4, 5, 6, 7). Please define this, i.e. Would all of those placentas come from the same dam? Or would these be from separate dams? The 'n' value should be the dams, not the placentas.

The n for figures listed are placentas. We always aim for at least 2 different litters/dams when analysing any molecular/cellular phenotype; however, these experiments were performed by student who left the lab long time ago and we cannot locate the exact information about how many pregnancies these placentas were obtained from. Thus, we describe those tissues came from at least 2 different dams.

WB for Flag was performed from 3-5 different dams/gestational time point.

6. What does the red arrow in Figure 1e represent? Please add this to the figure legend.

We have removed the arrow and updated the figure.

7. Figure 2 legend, "*Smpd1*^{-/-} fetuses exhibit reduced birth weights" is not a correct statement. Fetuses cannot have "birth" weights. Please reword.

Corrected.

8. Figure 3 - please indicate the *Spg* and *Lab* and *Dec* areas on the images in 3a.

We will add the labelling as suggested.

9. The statement on Page 9, beginning with "Given that *Smpd1* regulates the production of S1P..." does not take into account the many other enzymes involved in both production and degradation of S1P downstream of *Smpd1* activity. Please revise this statement to reflect this.

Statement was changed to: *Smpd1* is one of the enzymes involved in the production of S1P.

10. Please expand on the Lipidomic Profile methods used and provide a relevant reference. More details are needed for the method. How many dams? How many placentas per dams? What is meant by fold change compared to WT? How was this done? Were all WT placentas averaged and this used to do the fold change?

More details are provided in the revised manuscript. 3 WT and 3 KO placentas from 2 different litters were analysed individually and obtained values were compared between genotypes.

Reviewer3.

1. It is unclear if the *Smpd1* flag/flag mouse is an appropriate model to delineate the spatial expression pattern of SMPD1 in the placenta, considering that a prominent non-specific band showed on the Western blot. The low magnification image in Suppl. Fig. 1 does not allow to determine if there is also signal in the WT control. Higher magnification images would be required to exclude non-specific signals. Fig. 1e should include WT controls.

We have included these images in Figure 1D, Control refers to WT placenta exposed to Flag Ab and the same concentration as Smpd1 flag/flag tissue below.

Our search indicated that polyclonal anti-SMPD1 antibodies are available and suitable for IHC.

While there are numerous Smpd1 Ab available; their specificity is questionable. Immunoreactivity tested for 2 Abs we purchased was proven non-specific (Proteintech 14609-1-AP, and Novus NBP2-20426- see figure of WB). We also tested these Abs on immunostaining using paraffin embedding with similar outcomes. We have also tested Smpd1 Ab (Bioss, bs-6318R) on IHC and this antibody did not produce any immunoreactivity, despite companies' recommendation for use of this AB for IHC-P.

2. While the study focuses on cells of the trophoblast lineage in the fetal labyrinth, the potential contribution of the junctional zone (JZ) trophoblast is not investigated. This is important since the JZ size is increased in the LOF mice. Note that the JZ contains both, spongiotrophoblast and glycogen trophoblast, the latter of them contain glycogen storage vesicles. It will be important to determine if SpT, GlyT or both cell types are more abundant in the mutant JZ.

We have analysed cell distribution within junctional zone, and we did not find significant differences in the spongio and glycogen cell compartment.

3. It is unclear if Smpd1 is expressed in fetal endothelial cells of the labyrinth, and whether its LOF results in vascular phenotypes in the placenta. This is particularly relevant, since the authors state that S1P acts downstream of Smpd1 in the sphingolipid signaling pathway and is important for promoting angiogenesis. To address this, authors should perform co- staining with an endothelial marker, e.g. CD31/PECAM.

Thank you for the suggestion. We have attempted to do this experiment; however, because the smpd1 Ab is not specific, we could not unequivocally establish whether endothelial cells express smpd1.

4. In addition to the point #3, CD31 / PECAM staining will allow to determine the size of maternal and fetal blood space in the LOF placentas. Figure 7b suggests that both, maternal and fetal blood spaces are reduced, but this should be quantified.

We have analyzed fetal arterial tree and we did not find significant differences in the surface area, volume, diameter distribution, branching or span of fetal placental vessels. However, the depth to which the vasculature grew was significantly shallower. Data are included in FigS2.

As suggested, we also quantitated maternal blood spaces and we observed significant reduction in KO labyrinth both at day 15.5. and 17.5. (Fig 3D).

5. Please clarify if WT controls shown in Fig. 2 are in the same mixed genetic background as the LOF mutant mice? Otherwise, the differences shown here could be due to strain-specific effects on the fetal weights.

Yes, we confirm that all data generated used WT from the same mixed genetic background bred alongside our Asm KO colony.

6. The authors show in Figure 4 decreased Sphingosine kinase activity and decrease S1P activity in mutant placentas. The authors should discuss the discrepancy between human studies correlating high circulating S1P levels with pregnancy disorders including IUGR and their mouse model with reduced S1P levels and the IUGR phenotype.

Our data reflect tissue (placental) levels of S1P. We have not analysed circulating S1P levels. Circulating levels vs tissue levels are not the same since different cell types contribute to the circulating S1P pool (platelets, vasculature, etc.). Consistent with our findings in Smpd1 mutant placentas, del Gaudio et al. 2020 reported increased levels of Sphingomyelin and trend towards decreased levels of S1P in placental chorionic arteries isolated from preeclamptic placentas. (1)

7. Fig. 7: without staining for SpT or GyT specific markers, the authors cannot conclude that the JZ invaginations shown in panel 7b are SpT.

We agree and we have removed this statement from the text.

8. In their discussion of the results, and also in the abstract, the authors should state the hypothesis that the decreased size of the labyrinth leads to a decreased area of exchange between maternal and fetal blood space, thus limiting supply of

nutrients to the fetus and resulting in IUGR.

Thank you for the suggestion. We have incorporated this statement in the text.

Minor comments

1. All Figures: add error bars to the graphs - datapoints and errors bars have been added
2. Fig. 2a: label images of embryos in the box next to the graph. Labels were included
3. Fig. 3: labelling of panels does not align with the panels. Corrected
4. State in M&M if placental tissue used for Westerns, lipidomics profiles, and qRT-PCR contain decidua of just the fetal part of the placenta. All tissues contain decidua. This was included in a text.
5. Fig.4: provide in the figure legend the full names of the enzymes shown in Fig. 4c. This was added.
6. Authors should replace "interhemal membrane" in the text with "interhaemal distance" as used in the abstract. This was corrected.

All requested changes have been incorporated into the revised manuscript.

Reference

1. Del Gaudio I, Sasset L, Lorenzo AD, Wadsack C. Sphingolipid Signature of Human Feto- Placental Vasculature in Preeclampsia. *Int J Mol Sci.* 2020;21(3).

Second decision letter

MS ID#: dev.204425R1

MS TITLE: Acid sphingomyelinase is a gatekeeper of placental labyrinthine architecture and function.

AUTHORS: Andrea Jurisicova, Isidora Rovic, Katherine Szelag, Han Li, Rosanne McQuaid, Sara Sugin, John Sled and Natalia Theodora

Dear Dr Jurisicova,

I have now received all the referees reports on the above manuscript, and have reached a decision. The referees' comments are appended below, or you can access them online: please go to .

The overall evaluation is positive and we would like to publish a revised manuscript in *Development*, provided that the referees' comments can be satisfactorily addressed. Please attend to all of the reviewers' comments in your revised manuscript and detail them in your point-by-point response. If you do not agree with any of their criticisms or suggestions explain clearly why this is so. If it would be helpful, you are welcome to contact us to discuss your revision in greater detail. Please send us a point-by-point response indicating your plans for addressing the referees' comments, and we will look over this and provide further guidance.

Reviewer 2

Advance summary and potential significance to field

The primary objective of this study is to investigate whether sphingomyelin-phosphodiesterase 1 (Smpd1) is required for normal placental development in the mouse. The authors found that deficiency of Smpd1 results in intrauterine growth restriction (IUGR) was associated with reduced placental labyrinthine compartments and increased fetal-maternal interhaemal distance. Sphingosine 1-phosphate (S1P), produced downstream of Smpd1, is reduced three-fold. The placentas from Smpd1 deficient mice also show defective autophagy and lysosomal impairment.

Comments for the author

The authors have spent considerable effort to revise their manuscript according to the reviewer's comments and suggestions.

Some additional comments and suggestions:

1. Fig S1-A - This figure now shows consistent staining in the junctional and in the labyrinthine zones of the placenta. However, there is also now considerable staining of what look to be the maternal components (decidua, mesometrial). Are these components expected to express SMPD1? With this type of staining, can the authors still say that there was enrichment in the spongiotrophoblast and labyrinth zones (figure legend for Fig S1-A)? In the results text, the authors indicate that "strong immunoreactivity was present in sinusoidal trophoblast giant cells (sTGC) within the labyrinth". However, looking at Fig S1-A, everything looks strongly positive. If an emphasis is warranted for sTGC, then please use a pointer to indicate these cell types compared to others.
2. I appreciate the clarification that most results are based on the number of placentas, not the number of dams and that placentas from at least two dams were analyzed for each analysis. However, it is important to include this information in your manuscript. Please include this last part about the number of dams in each appropriate figure legend. This is important to provide context for interpretation of the robustness of the results.
3. Similar to labels for Fig 3A, please put labels on Supplemental Figure S1-A. I would also suggest identifying the junctional zone in each of these figures.
4. Suggestion to include in the discussion the authors response to Reviewer 3's questions about the discrepancy in results with S1P levels and IUGR. This is an important point and is not discussed.

Reviewer 3*Advance summary and potential significance to field*

The authors uncover novel functions for Smpd1, a key enzyme of the sphingolipid metabolic pathway, during placental development in mice. The revised manuscripts has significantly improved the overall strong experimental findings and should now be accepted for publication.

Comments for the author

Highlighting the changes in the revised manuscript would make it easier for the reviewer to located to edits.

Second revisionAuthor response to reviewers' comments

Dear D, Hadjantonakis.

We would like to thank you and reviewers for re-evaluation and feedback how to improve the manuscript. We appreciate the opportunity to address the comments and have made a small modifications to the manuscript. Details are described below.

1. Fig S1-A - This figure now shows consistent staining in the junctional and in the labyrinthine zones of the placenta. However, there is also now considerable staining of what look to be the maternal components (decidua, mesometrial). Are these components expected to express SMPD1? With this type of staining, can the authors still say that there was enrichment in the spongiotrophoblast and labyrinth zones (figure legend for Fig S1-A)? In the results text, the authors indicate that "strong immunoreactivity was present in sinusoidal trophoblast giant cells (sTGC) within the labyrinth". However, looking at Fig S1-A, everything looks strongly positive. If an emphasis is warranted for sTGC, then please use a pointer to indicate these cell types compared to others.

Response: The Smpd1-Flag staining. We have further optimized the staining using Proteintech Ab (new retrieval method) and we included the outcomes of these studies in figure 1D. Antibody (Rockland) specifically stains subset of sinusoidal and canal giant cells with occasional patchy staining in spongiotrophoblast. Cells are indicated in the image Fig 1D with arrowhead, star and arrow respectively.

2. I appreciate the clarification that most results are based on the number of placentas, not the number of dams and that placentas from at least two dams were analyzed for each analysis. However, it is important to include this information in your manuscript. Please include this last part about the number of dams in each appropriate figure legend. This is important to provide context for interpretation of the robustness of the results.

Response: Number of placentas/dams has been indicated in each fig legends.

3. Similar to labels for Fig 3A, please put labels on Supplemental Figure S1-A. I would also suggest identifying the junctional zone in each of these figures.

Response: Labels for individual zones were included in the figures 3 and 6.

4. Suggestion to include in the discussion the authors response to Reviewer 3's questions about the discrepancy in results with S1P levels and IUGR. This is an important point and is not discussed.

Response:

Discussion about circulating vs tissue based S1P levels us now included in the discussion section with appropriate citation included.

All changes described in the response have been highlighted in the manuscript.

Thank you for your advice and guidance during the preparation of this manuscript.

Andrea

Third decision letter

MS ID#: dev.204425R2

MS TITLE: Acid sphingomyelinase is a gatekeeper of placental labyrinthine architecture and function.

AUTHORS: Andrea Jurisicova, Isidora Rovic, Katherine Szelag, Han Li, Rosanne McQuaid, Sara Sugin, John Sled, Natalia Theodora and Que Wu

Dear Dr Jurisicova,

I am happy to tell you that your manuscript has been accepted for publication in Development, pending our standard publication integrity checks.

Reviewer 2

Advance summary and potential significance to field

Comments for the author

The reviewers have been responsive to the suggestions. Thank you.

Reviewer 3

Advance summary and potential significance to field

The authors uncover novel functions for Smpd1, a key enzyme of the sphingolipid metabolic pathway, during placental development in mice. The revised manuscripts has significantly improved the overall strong experimental findings and should now be accepted for publication.

Comments for the author

As suggested by Reviewer 2, you have added the number of dams in the legends of Figs. 1, 3 and 5 . Please also include the number of placentas, either in the the Figure legends or at the bottom of the graphs.